# Community-Acquired *Legionella pneumophila* Pneumonia: A Case-Control Study in Adult Inpatients from 2019 to 2024

**DOI:** 10.3390/pathogens15010013

**Published:** 2025-12-22

**Authors:** Paola Di Carlo, Nicola Serra, Teresa Maria Assunta Fasciana, Francesca Fasciana, Luca Pipitò, Anna Giammanco, Angela Capuano, Caterina Carollo, Valentina Caputo, Tommaso Vincenzo Bartolotta, Consolato Maria Sergi, Antonio Cascio

**Affiliations:** 1Department of Health Promotion, Mother and Child Care, Internal Medicine and Medical Specialties, University of Palermo, 90127 Palermo, Italy; paola.dicarlo@unipa.it (P.D.C.); teresa.fasciana@unipa.it (T.M.A.F.); lucapipito@gmail.com (L.P.); caterina.carollo@unipa.it (C.C.); valentina.caputo@unipa.it (V.C.); antonio.cascio03@unipa.it (A.C.); 2Department of Neuroscience, Reproductive Sciences and Dentistry, Audiology Section, University of Naples Federico II, Via Pansini 5, 80131 Naples, Italy; 3Degree Course in Medicine and Surgery, University of Palermo, 93100 Caltanissetta, Italy; francesca.fasciana@gmail.com (F.F.); anna.giammanco@unipa.it (A.G.); 4AORN Santobono-Pausilipon, Healthcare Directorate, 80128 Naples, Italy; a.capuano@santobonopausilipon.it; 5Department of Radiology, University Hospital “Paolo Giaccone”, 90127 Palermo, Italy; tommasovincenzo.bartolotta@unipa.it; 6Anatomic Pathology Division, Children’s Hospital of Eastern Ontario, University of Ottawa, Ottawa, ON K1H 8M5, Canada; csergi@cheo.on.ca; 7Department of Laboratory Medicine and Pathology, University of Alberta, Edmonton, AB T6G 2R3, Canada

**Keywords:** *Legionella pneumophila*, hospitalized patients, clinical, blood parameters

## Abstract

Background: *Legionella pneumophila* is frequently acquired in the community and generally linked to contaminated domestic hot water systems, hotels, or other environmental sources or of unknown origin. *L. pneumophila* is a leading cause of pneumonia, especially in high-risk patients such as those over 50 who are immunocompromised or people with pre-existing illnesses. This study examines the factors linked to *L. pneumophila* acquired in the community in adult patients with hospitalization due to pneumonitis. Methods: This case-control study included 140 hospitalized adult patients admitted to the University Hospital Paolo Giaccone of Palermo between January 2019 and December 2024. Demographic, laboratory, clinical, and microbiological data were collected electronically. Urinary antigen testing and the BioFire FilmArray Pneumonia Panel were used to detect *L. pneumophila* and were performed within 48 h upon admission. Results: Of 140 pneumonia patients, 70 were positive (SG) and 70 were negative (CG) for *L. pneumophila*. Comorbidities were significantly associated with the presence of *L. pneumophila* (*p* = 0.0046). The most frequent comorbidity was only heart failure (*p* < 0.0015) and, similarly, for smoke (*p* = 0.0487). There was no difference in mortality between the two groups (SG). Levofloxacin was the most frequent therapy used in SG (*p* < 0.0001). Additionally, SG showed significantly lower blood sodium, phosphate, and platelet levels (all *p* < 0.0001) compared to the CG. In contrast, blood parameters such as LDH, CRP, AST, and ALT were significantly higher (all *p* < 0.0001). Conclusions: Our research highlights the critical need for early detection of *L. pneumophila* infections, especially in patients with high CRP levels, moderate hypophosphatemia, or heart failure. In these patients with *L. pneumophila*, early treatment with macrolide and fluoroquinolone is mandatory to reduce mortality.

## 1. Introduction

*Legionella* infections are a global concern. The Global Burden of Disease 2021 (GBD) study, which assessed 371 diseases and injuries, provided new data on *Legionella* spp.—covering incidence, prevalence, years lived with disability (YLD), and disability-adjusted life years (DALYs). Despite efforts to develop a unified worldwide strategy to lower its public health impact, marked differences persist by region and age [1,2,3].

A thorough understanding of *Legionella*’s burden and epidemiological patterns is vital to effectively cutting worldwide mortality and disease rates by 2030 [1]. Legionnaires’ disease (LD) is a systemic infectious disease primarily involving the lungs, with multisystemic extrapulmonary manifestations [4]. In particular, LD occurs when people inhale *Legionella* bacteria from water or soil aerosols. Individuals over 50, those who are immunocompromised, or people with pre-existing illnesses face higher risks [2,5,6].

*L. pneumophila* community-acquired pneumonia, which characteristically targets individuals older than 50 years, is becoming increasingly common across Italy, probably due to the aging population and low natality rate [7,8]. Recent surveillance data indicate an apparent rise in incidence rates, with Italy experiencing its highest recorded annual rate of Legionnaires’ Disease—a 25% increase from 2022 [7]. This increasing prevalence aligns with similar trends observed across Europe, where reported legionellosis cases have risen considerably in recent years, prompting public health concerns [9,10]. Notably, a substantial majority of these infections are acquired in community settings rather than in healthcare facilities [11,12].

*Legionella* spp. outbreaks in Italy are occurring not only in hospital settings [13] but also in high-risk communities, including prisons [14]. This highlights the importance of monitoring and addressing potential sources of infection in diverse environments [15,16,17].

For this reason, a regional surveillance system has been established in Sicily, which includes the University Hospital in Palermo, Italy [18,19,20].

Environmental factors—such as increased precipitation and sustained humidity—are believed to amplify further the risk and incidence of *L. pneumonitis*, particularly among older adults. Such conditions create favorable environments for *Legionella* proliferation, thereby increasing the risk of exposure for susceptible populations. Therefore, continued monitoring and public health interventions are essential to mitigate this rising public health threat in the Mediterranean region [21,22,23,24].

Elderly individuals, particularly those aged 60 years and above, exhibit a heightened vulnerability to severe manifestations of the disease, frequently necessitating hospitalization, especially in the presence of pre-existing health conditions such as diabetes, chronic obstructive pulmonary disease, or cardiovascular disorders. In Italy, research indicates that the median age of patients hospitalized due to *Legionnaires*’ disease typically ranges between 63 and 69 years, underscoring the burden of this infection among older populations. Many of these patients present with multiple comorbidities, which may exacerbate their clinical outcomes [25,26]. Older age, combined with existing health issues, is widely recognized as a significant risk factor that increases both the severity of illness and the mortality rates linked to *Legionella* infections. Numerous national and international studies support this claim, highlighting the critical relationship between age and overall health in these cases [25,26,27].

Diagnosis is often missed because patients rarely receive specific tests before antibiotics are given, and diagnosis relies mainly on the urinary soluble antigen test. Most available tests detect only *L. pneumophila* serogroup 1, leading to underdiagnosis of other serogroups across Europe [27,28]. Regional differences in reporting also result in variable incidence rates across Italy [13,24,29].

European hospitals emphasize the importance of early diagnosis in managing pneumonia cases, particularly when *Legionella* infection is suspected. Many studies have sought to identify the unique clinical, radiological, and laboratory features that distinguish Legionnaires’ disease from other types of community-acquired pneumonia [30]. Pneumonitis caused by *L. pneumophila* typically presents with nonspecific clinical symptoms such as fever, cough, and dyspnea, and extrapulmonary signs such as gastrointestinal symptoms (diarrhea, nausea), confusion, and hyponatremia. Previous papers have proposed a rapid diagnostic system based on a weighted scoring method that incorporates key clinical and laboratory data. By emphasizing extrapulmonary organ involvement and laboratory abnormalities, clinicians can make more rapid presumptive diagnoses. This approach improves diagnostic specificity, enabling timely and accurate disease identification [31,32,33,34,35].

Radiologically, the findings can include patchy or lobar infiltrates on chest X-rays, which are often nonspecific and can mimic other types of pneumonia [28,30].

The main challenges in diagnosing *Legionella* infection in hospitalized patients using rapid, low-cost methods are limited sensitivity and specificity, limited detection of species and serogroups, and specimen acquisition issues.

The most widely used rapid and low-cost test is the urinary antigen test (UAT), which primarily detects *L. pneumophila* serogroup 1 (Lp1) [36]. This test misses infections caused by other serogroups and species, which are more common in immunocompromised and nosocomial cases, leading to underdiagnosis in these populations. Sensitivity of UAT for Lp1 ranges from 56 to 99% but can be as low as 40% for non-Lp1 strains or in immunosuppressed patients. Additionally, UAT does not provide isolates for epidemiological typing or antimicrobial susceptibility testing, which is critical for outbreak investigations and public health management [13,37,38]. Regarding lower respiratory tract samples (e.g., sputum or BAL), many rapid biomolecular panels primarily detect *L. pneumophila*, often limited to serogroup 1. They may miss infections caused by other *Legionella* spp. or serogroups. This is a significant limitation, as non-pneumophila species and non-sg1 serogroups can cause disease, particularly in nosocomial settings [27,32,33]. There is currently no universally accepted, detailed protocol for *L. pneumophila* antimicrobial susceptibility testing (L-AST), which prevents the establishment of resistance standards. This lack of standardization is primarily due to challenges with traditional L-AST methods that use media containing activated charcoal, which can absorb antibiotics and yield inaccurate results. Research is ongoing to establish a standardized broth microdilution (BMD) method as the future gold standard for L-AST, which is a critical step toward defining resistance boundaries [39]. Therefore, treatment for Legionnaires’ disease is primarily based on empirical guidelines using antibiotics known to achieve high intracellular concentrations (e.g., fluoroquinolones and macrolides), as *Legionella* is an intracellular pathogen and beta-lactams (which often have established MIC breakpoints) are ineffective despite potential in vitro activity. Delayed antibiotic treatment is associated with a worse prognosis, so treatment is initiated quickly without waiting for susceptibility testing [38].

In our quality assurance (QA)-certified reference laboratory for the environmental surveillance program of *Legionella* spp., legionellosis surveillance in Sicily is part of a national framework coordinated by the Istituto Superiore di Sanità and the Ministry of Health, with regional implementation through acts of the Department of Health and the Regional Commission for Legionellosis [17,18,19]. All hospitalized patients admitted for severe pneumococcal pneumonia undergo *Legionella* urinary antigen testing according to international [1,6,36,37,38] and Italian recommendations for early diagnosis and treatment of *L. pneumophila* [7,13,24,40].

This study investigates whether early diagnosis and treatment of *L. pneumophila* improves outcomes in adults hospitalized with community-acquired pneumonia, including patients requiring intensive care. It also identifies key factors related to *L. pneumophila* in these patients.

## 2. Materials and Methods

### 2.1. Study Design

This case-control study was conducted from January 2019 to December 2024 at the Policlinico University Hospital Paolo Giaccone (A.U.O.P.), University of Palermo, Italy.

In this study, the patients were stratified into control and study groups (CG and SG, respectively), consisting of those ≥18 years or older with community-acquired pneumonia who were negative or positive for *L. pneumophila*. Both the control and study groups were composed of 70 hospitalized patients who underwent urinary antigen testing, respiratory panels, and BioFire FilmArray™ PNplus Panels (BioFire Diagnostics, Salt Lake City, UT, USA) performed within 48 h upon admission. To reduce statistical bias, both groups were randomly selected and were comparable in size, age, sex, and time period. In particular, regarding the period, we enrolled the same number of patients for each year and for each season. Particularly, we excluded from this study all patients positive for HIV and SARS-CoV-2 infection. All patients were admitted to the emergency department of A.U.O.P. and defined as having a new or progressive existing chest X-ray infiltrate, in addition to a clinical manifestation suggesting a respiratory infection (fever and respiratory symptoms) as previously reported [41]. Data on demographics, underlying diseases, laboratory parameters, pathogens, treatment, and outcomes were retrospectively collected from the electronic records.

Chronic lung disease comorbidities included chronic obstructive pulmonary disease, interstitial lung disease, bronchiectasis, emphysema, asthma, and chronic bronchitis.

Additionally, heart failure comorbidity included all patients, according to Bozkurt B. et al. [42].

The authors classify the medical setting in three categories—medicine, surgery, and intensive care unit (ICU)—reflecting differences in procedures and patient populations. Particularly, the medicine area included cardiology, endocrinology, gastroenterology, geriatrics, hematology, infectious diseases, internal medicine, nephrology, neurology, oncology, and pulmonology wards. The surgery area included cardiovascular surgery, general surgery, neurosurgery, oncological surgery, otolaryngology, orthopedic surgery, plastic surgery, urology, vascular surgery, and obstetrics and gynecology. Finally, the intensive care unit included general intensive care, postoperative intensive care, and specialized intensive care.

Urinary samples, sputum, and bronchoalveolar lavages (BALs) were obtained from all enrolled patients. Urine samples were collected to determine the presence of *L. pneumophila Sg 1–15 L. longbeachae* on urine detection by DiaSorin S.p.A., (Saluggia, Italy), as previously reported [20]. 

Bacterial and fungal pathogens using standard sputum culture and/or BioFire FilmArray Pneumonia Panel Plus, which can identify 15 typical bacteria, three atypical bacteria, and seven antimicrobial resistance (AMR) genes. For *Chlamydia pneumoniae*, *Mycoplasma pneumoniae*, and *L. pneumophila*.

Antibiotic treatment for *L. pneumophila* was defined as the use of macrolides or fluoroquinolones. The macrolide used in this study was azithromycin. The fluoroquinolone used was levofloxacin. Particularly, levofloxacin was preferred to other fluoroquinolones since it was recommended as monotherapy in the initial treatment of community-acquired pneumonia patients, as suggested by Metlay et al. [43].

### 2.2. Statistical Analysis

Data were presented as numbers and percentages for categorical variables, and continuous data were expressed as the mean and standard deviation (SD) or median and interquartile range (IQR = [Q1, Q3]). The chi-square test (C) and Fisher’s exact test (F) were performed to evaluate significant differences in proportions or percentages between the two groups. The multiple comparison chi-square and Fisher’s exact test were used to define significant differences among three or more percentages for unpaired data. If the chi-square or Fisher’s exact test were significant (*p*-value < 0.05), the post hoc test was performed using the adjusted standardized residuals and the z-test (Z). Fisher’s exact test was used where the chi-square test was not appropriate.

The normality test was performed using the Shapiro–Wilk test. The Shapiro–Wilk test is a statistical test used to determine if a sample of data comes from a normally distributed population. It compares the sample data distribution to a perfect normal distribution, producing a W statistic and a *p*-value. A *p*-value greater than the significance level (e.g., 0.05) suggests the data is normally distributed, while a *p*-value less than the significance level indicates it is not. A Shapiro–Wilk test is used to determine if a dataset is normally distributed, while a *t*-test is used to compare the means of two groups. The Shapiro–Wilk test is often performed as a preliminary step before a *t*-test to check if the normality assumption for the *t*-test is met.

The *t*-test was used to evaluate the differences between two means of unpaired data. Alternatively, the Mann–Whitney test was used if the data distributions were not normal.

Multiple logistic regression was used to identify the best-fit model for individualizing the significant predictors among the laboratory parameters considered in the bivariate analysis. For this step, we considered the dependent variable *Legionella*_var, which assigns 1 to patients positive for *L. pneumophila* and 0 to those negative.

Finally, all tests with a *p*-value (*p*) < 0.05 were considered significant. The statistical analysis was performed using the MATLAB (Matrix Laboratory) analytical toolbox R2008b (MathWorks, Natick, MA, USA).

## 3. Results

The control group (CG) was composed of 74.3% (52) males and 25.7% (18) females, with ages ranging from 18 to 90 years, a mean of 64.3 years, and a standard deviation of 17.8 years. The study group (SG) comprised 62.9% (44) males and 37.1% (26) females, with ages ranging from 26 to 89 years, a mean of 66.7 years, and a standard deviation of 15.5 years. In Table 1, we reported the characteristics for both groups. In the last column the comparison was performed between them.

Table 1 shows no significant differences between CG and SG for age (*p* = 0.50), gender (*p* = 0.70), hospitalization days (*p* = 0.43), and medical area (*p* = 0.25). Additionally, we found more frequent use of oxygen therapy in patients negative to *L. pneumophila* (80.0% vs. 61.4%, *p* = 0.0158), while the risk factor of smoking was more present in patients positive to *L. pneumophila* (82.9% vs. 78.6%, *p* = 0.0487). A relationship between the commodity type identified in this study and patients positive to *L. pneumophila* was found (*p* = 0.0018). Notably, in patients positive for *L. pneumophila*, a higher proportion of patients with heart failure (37.1% vs. 5.7%, *p* = 0.0015) was observed. Regarding therapy for *L. pneumophila* doxycycline was more used in patients negative to *L. pneumophila* (67.1%, *p* < 0.0001), while levofloxacin was more used in patients positive to *L. pneumophila* (61.4%, *p* < 0.0001). In Table 2, we report the values of laboratory parameters, including sodium, ipofosfatemia, LDH, CPR, platelets, AST, and ALT, for CG and SG.

From Table 2, between CG and SG, we found a significant reduction for sodium (median: 139 vs. 134, *p* < 0.0001), hypophosphatemia (median: 3.6 vs. 2.9, *p* < 0.0001), and platelets (median: 2.3 × 10^5^ vs. 1.4 × 10^5^, *p* < 0.0001); a significant increase was observed for LDH (median: 235.0 vs. 443.5, *p* < 0.0001), CRP (median: 34.0 vs. 262.5, *p* < 0.0001), AST (median: 26.5 vs. 70.5, *p* < 0.0001), and ALT (median: 34.5 vs. 48.5, *p* < 0.0001).

Table 3 shows the multiple logistic regression between *Legionella_var* and the significant laboratory parameters described in Table 2, CRP was a significant positive predictor for positive patients with *L. pneumophila* (OR = 1.01, *p* < 0.0001). At the same time, hypophosphatemia was a significant negative predictor of positive patients for *L. pneumophila* (OR = 0.15, *p* = 0.0027); i.e., high CRP values or low hypophosphatemia levels were associated with a higher probability of positive patients for *L. pneumophila*.

## 4. Discussion

This study highlights patient characteristics and laboratory differences between hospitalized patients with community-acquired pneumonia who tested positive or negative for *L. pneumophila*.

Although demographic characteristics and hospitalization duration were similar between the two groups, risk factors, such as smoking and laboratory parameters (Table 2) were significantly associated with *Legionella*-infection-positive cases.

Regarding the mortality, no significant differences were found between CG and SG (25.7% vs. 20.0%, *p* = 0.65). Notably, the percentage of mortality in SG was in line with the results obtained in other studies such as Dagan A. et al., Graham F. et al., and Pessoa, E. et al. [44,45,46], which reported similar results for patients with age greater than 65 years and with concomitant pathologies, mainly of the chronic–degenerative type (diabetes, hypertension, chronic-obstructive bronchopathy), neoplastic, autoimmune, infectious, transplants, and other pathologies.

In our study, tobacco smoking is a significant risk factor in hospitalized patients with pneumonia due to *L. pneumophila*. Worldwide studies show that both current and former smokers have a much higher risk than non-smokers [47]. Cigarette smoke weakens lung defenses by reducing the number of alveolar macrophages needed to clear *Legionella*. Animal models confirm that smoke exposure increases bacterial load and disease severity by impairing immunity. Additionally, the pro-inflammatory and toxic effects of smoking further damage lung clearance and tissue integrity [48]. Among our enrolled patients, smoking habits reflect national trends noted by the National Institute of Health: From 2022 to 2024, smoking increased in Southern Italy and the Islands, especially among those aged 50–69 [49].

This study identified, among comorbidities, a significantly higher incidence of heart failure in patients with *L. pneumophila* (*p* < 0.0015). Patients with pre-existing heart disease, including heart failure, are at higher risk for severe outcomes from Legionnaires’ disease. Studies identify chronic cardiovascular disease as an independent risk factor for both contracting *Legionella* and experiencing more serious illness, such as ICU admission and increased mortality [25,50]. However, there is no evidence of a direct mechanistic link between *Legionella* infection and the development or progression of heart failure. Recent studies highlight the need to investigate long-term effects of *Legionnaires*’ disease, particularly persistent symptoms and quality of life, but do not specifically address heart failure. While cardiac complications and increased risk for heart failure patients are recognized, no research directly examines their link to *Legionella* [25,26,50]. When evaluating the impact of heart failure on Italian patients admitted to hospital for legionellosis, it is essential to consider the Italian demographic context. This involves examining how the age distribution of the Italian population compares with that of other European nations. Italy is known for having an older population, which may influence both the prevalence of heart failure and the outcomes of *Legionella* infections [23].

Oxygen therapy was administered more frequently in the control group. In our study, we used urinary antigen, which does not discriminate the serogroup of *Legionella*. For this reason, the use of fluoroquinolones such as levofloxacin can be considered advantageous, especially in patients with heart failure and *L. pneumophila* infection. Multiple studies have demonstrated that levofloxacin exhibits potent in vitro activity against both serogroup 1 and non-serogroup 1 isolates of *L. pneumophila*, as well as other *Legionella* species, with no significant differences in minimum inhibitory concentrations (MICs) or evidence of resistance among environmental or clinical isolates [51].

In vitro studies confirm that levofloxacin and macrolides (azithromycin and clarithromycin) achieve adequate intracellular concentrations and are highly active against *Legionella* spp. [51,52].

There are rare reports of reduced susceptibility to macrolides in *L. pneumophila* isolates in Italy, primarily associated with the presence of the lpeAB efflux pump gene, but clinical resistance remains uncommon. Studies on environmental and clinical isolates from different Italian regions showed that most *Legionella* strains remain susceptible to both macrolides and fluoroquinolones, with no confirmed cases of high-level fluoroquinolone resistance and only isolated cases of reduced macrolide susceptibility [PD2] [53]. However, we must remain vigilant for the possible emergence of serogroups resistant to anti-*Legionella* treatments, as reported by Cruz et al. [54].

Given the lack of consensus on a validated diagnostic scoring system [54], we focused our investigation on standard laboratory parameters. With this investigation, we aimed to identify possible variability of laboratory parameters linked to *L. pneumophila*. Particularly, the patients with *L. pneumophila* compared to non-*Legionella* pneumonia cases showed lower sodium blood levels, blood phosphate levels, and platelet levels. Vice versa, higher values of LDH, CRP, AST, and ALT were observed in patients with *L. pneumophila*.

Logistic regression analysis identified CRP as a positive predictor and hypophosphatemia as a negative predictor for *L. pneumophila* positivity, suggesting that elevated inflammatory response and low phosphate levels may be helpful to clinical indicators *L. pneumophila* presence [55,56].

In our study, we observed that patients with *L. pneumophila* infection showed a more pronounced increase in CRP levels than those with pneumonia caused by other etiological agents. This marker elevation in CRP was a notable finding and suggests a potentially distinct inflammatory response associated with *Legionella* infections.

This laboratory finding is in accordance with other studies [36,57]. In particular, a recent study by Klopfenstein T. et al. [36] showed that a C-reactive protein (CRP) threshold below 130 mg/L has a strong negative predictive value for excluding *L. pneumophila* serogroup 1 infection.

Additionally, we found a low-significant presence of hypophosphatemia in patients positive for *L. pneumophila*. These values were about 2.8, compared to 3.4 detected in patients without *L. pneumophila*. Despite the lower levels, these are not dangerous. We suppose that the lower levels of hypophosphatemia in the study group (SG) are due to the significantly high presence of patients with heart failure [58]. Similarly, for the sodium blood level, we found lower but not dangerous levels between the control and study groups (138.4 vs. 134.9, *p* < 0.0001), also due to the high prevalence of patients with heart failure [19].

High transaminase levels, specifically aspartate aminotransferase (AST) and alanine aminotransferase (ALT), are frequently observed in patients with *L. pneumophila* [59], according to our results.

The authors recommend that laboratory parameters significantly associated with *L. pneumophila* infection can serve as indicators of possible *L. pneumophila* infection, but that they should be confirmed by microbiological analyses, according to the guidelines of the American Thoracic Society and the Infectious Diseases Society of America [60,61].

Future surveillance of resistance or other *Legionella* spp. in hospitalized patients should include routine environmental monitoring of hospital water systems for *Legionella* spp, combined with clinical surveillance of patients at risk for nosocomial pneumonia [23,62]. Environmental surveillance should utilize both culture-based and molecular methods to detect a broad range of *Legionella* spp. and serogroups, not just *L. pneumophila* serogroup 1, as non-*pneumophila* and non-serogroup 1 species are increasingly recognized as causes of disease [28].

Our findings underscore the importance of early consideration of *L. pneumophila* infections, particularly in patients presenting with elevated protein C levels, moderate hypophosphatemia, or underlying heart failure. Timely, low-cost, and effective microbiological investigations, along with targeted antibiotic therapy with levofloxacin, can markedly improve clinical outcomes and mitigate the burden of Legionnaires’ disease among hospitalized patients with *L. pneumophila*-associated pneumonitis. Adopting a comprehensive, multidisciplinary approach to predictive diagnostics can further optimize the management of *L. pneumophila* infection in hospital settings. Additionally, sustained environmental surveillance and ongoing monitoring of antibiotic resistance are crucial measures to reduce mortality attributable to *L. pneumophila* in healthcare settings.

### Limitations

This single-center study provides preliminary clinical data on hospitalized patients with pneumonitis caused by *L. pneumophila*. The study focused on low-cost, effective diagnostic strategies and assessed the links between *L. pneumophila* infection, smoking habits, patient characteristics, and laboratory results. Despite a specific geographical area, such as the Sicilian region, interesting associations were found with smoking, heart failure, and laboratory parameters.

In our study, we used urinary antigen, which does not discriminate the serogroup of *Legionella*. In fact, conventional urinary antigen assays are designed to detect only *L. pneumophila* serogroup 1 and do not provide information about other serogroups or species. This limitation means that infections caused by non-serogroup 1 strains or other *Legionella* spp. may be missed, and the test result cannot be used to identify the specific serogroup responsible for infection.

## 5. Conclusions

This study aligns with some previous research while diverging from others. In particular, our findings highlight the important role of heart failure as a comorbidity, as well as certain laboratory parameters such as hypophosphatemia and C-reactive protein (CRP). Smoking was confirmed as one of the most common risk factors among patients with *L. pneumophila*. Regarding therapy, levofloxacin was more frequently administered to patients with *L. pneumophila*. This result, together with the detection of *L. pneumophila* within 48 h at admission and the absence of significant differences in hospitalization time between the control group (CG) and the study group (SG), may explain the less frequent use of oxygen therapy in SG than in CG and both the non-significant need for intensive care and the lack of difference in hospital mortality between the two groups. These results may be due to the type of study design adopted. There are rarely case-control studies on community-acquired *L. pneumophila* in the scientific literature. Particularly, in Italy, only three recent studies were found. The published studies mainly concern retrospective studies on single groups of patients positive for community-acquired *L. pneumophila*, such as Falcone M. et al., (2021) [40], (116 pts) Lupia et al., (2023) [50] (50 pts), and Marino A et al., (2023) [24] (10 pts). Instead, our study is a case-control study characterized by control and study groups (CG and SG, respectively) composed of hospitalized patients with community-acquired pneumonia who were negative and positive for *L. pneumophila*, respectively. Both groups were composed of 70 patients (140 pts). The choice to consider this study design was to identify the parameters associated with *L. pneumophila* among patients with community-acquired pneumonia who required hospitalization, reducing statistical biases connected to other types of retrospective studies.

## Figures and Tables

**Table 1 pathogens-15-00013-t001:** Comparison between negative and positive patients to *L. pneumophila*.

Parameters	CG (Negative to *L. pneumophila*)	SG (Positive to *L. pneumophila*)	CG vs. SG *p*-Value (Test)
*Patients*	70	70	
*Age*			
Mean (SD)	64.3 (17.8)	66.7 (15.5)	
Median (IQR)	69.5 (50.0, 79.0)	70.0 (58.0, 78.0)	0.50 (MW)
*Gender*			
Males	74.3% (52)	62.9% (44)	0.70 (C)
Females	25.7% (18)	37.1% (26)	
*Hospitalization* (days)			
Mean (SD)	15.8 (12.2)	18.1 (14.0)	
Median (IQR)	13.0 (7.0, 21.0)	13.5 (9.0, 25.0)	0.43 (MW)
*Medical setting*			
Medicine	80.0% (56)	72.9% (51)	
Surgery	4.3% (3)	1.4% (1)	
Intensive Care Unit (ICU)	15.7% (11)	25.7% (18)	0.25 (F)
*Oxygen therapy*	80.0% (56)	61.4% (43)	0.0158 * (C)
*Non-invasive mechanical ventilation*	58.6% (41)	54.3% (38)	0.26 (C)
*Invasive mechanical ventilation*	27.1% (19)	24.3% (17)	0.15 (C)
*Mortality*	25.7% (18)	20.0% (14)	0.65 (C)
*Smoke*	78.6% (48)	82.9% (58)	0.0487 * (C)
*Comorbidities* †			
Diabetes	15.7% (11)	28.6% (20)	0.0046 * (F)
Heart failure	5.7% (4)	37.1% (26)	
*Chronic lung diseases*	61.4% (43)	67.1% (47)	Group B—heart failure **, *p* = 0.0015 (Z)
Hypertension	44.3% (31)	45.7% (32)	
Dialysis	10.0% (7)	27.1% (19)	
Liver cirrhosis	4.3% (3)	1.4% (1)	
Dementia	18.6% (13)	31.4% (22)	
*Sepsis*	11.4% (8)	22.9% (16)	0.073 (C)
*Cancer*	27.1% (19)	28.6% (20)	0.85 (C)
*Isolates* ††			
Gram−	15.7% (11)	27.1% (19)	
Gram+	5.7% (4)	10.0% (7)	0.44 (F)
Virus	10.0% (7)	7.1% (5)	
*Therapy for Legionella*			<0.0001 * (F)
no therapy ***	22.9% (16)	1.4% (1)	
doxycycline	67.1% (47)	35.7% (25)	Group A—doxycycline **, *p* < 0.0001 (Z)
levofloxacin	10.0% (7)	61.4% (43)	Group B—levofloxacin **,
azithromycin	0.0% (0)	1.4% (1)	*p* < 0.0001 (Z)

Note: † = some patients had more comorbidities, †† = some patients had more isolates, CG: negative to *L. pneumoniae,* SG: positive to *L. pneumoniae*, * = significant test; T = unpaired *t*-test; MW = Mann–Whitney test, C = chi-square test, F = Fisher’s exact test, ** = more frequent modality, Z = post hoc Z-test, *** = modality not included in the analysis.

**Table 2 pathogens-15-00013-t002:** Laboratory parameter comparison between negative and positive patients with *Legionella* infection.

Laboratory Parameters	CG (Negative to *L. pneumophila*)	SG (Positive to *L. pneumophila*)	CG vs. SG *p*-value (Test)
Sodium (mmol/L)			
Mean (SD)	138.4 (7.1)	134.9 (4.4)	
Median (IQR)	139.0 (136.0, 142.0)	134.0 (132.0, 139.0)	*p* < 0.0001 * (MW)
Ipofosfatemia (mg/dL)			
Mean (SD)	3.4 (0.6)	2.8 (0.5)	
Median (IQR)	3.6 (3.1, 3.8)	2.9 (2.4, 3.1)	*p* < 0.0001 * (MW)
LDH (U/L)			
Mean (SD)	290.0 (217.6)	465.5 (278.9)	
Median (IQR)	235.0 (199.0, 321.0)	443.5 (256.0, 567.0)	*p* < 0.0001 * (MW)
CRP (mg/dL)			
Mean (SD)	61.6 (72.8)	314.0 (362.5)	
Median (IQR)	34.0 (19.0, 69.0)	262.5 (135.5, 388.5)	*p* < 0.0001 * (MW)
Platelets			
Mean (SD)	2.2 × 10^5^ (0.8 × 10^5^)	1.5 × 10^5^ (1.1 × 10^5^)	
Median (IQR)	2.3 × 10^5^ (1.7 × 10^5^, 2.6 × 10^5^)	1.4 × 10^5^ (0.6 × 10^5^, 2.0 × 10^5^)	*p* < 0.0001 * (MW)
AST (U/L)			
Mean (SD)	54.7 (175.3)	105.8 (124.7)	
Median (IQR)	26.5 (23.0, 43.0)	70.5 (45.0, 103.0)	*p* < 0.0001 * (MW)
ALT (U/L)			
Mean (SD)	44.0 (70.2)	102.6 (157.3)	
Median (IQR)	34.5 (18.0, 45.0)	48.5 (37.0, 88.0)	*p* < 0.0001 * (MW)

Note: * significant test; T = unpaired *t*-test; MW = Mann–Whitney test.

**Table 3 pathogens-15-00013-t003:** Logistic regression analysis between *Legionella_var* and significant laboratory parameters described in Table 2.

Logistic Regression	Coefficient	Standard Error	OR	CI at 95%	*p*-Value
Null model vs. full model					<0.0001 * (C)
*Legionella_var*/Sodium	0.04	0.04	1.04	(0.95; 1.13)	0.38
*Legionella_var*/ipofosfatemia	−1.92	0.64	0.15	(0.04; 0.51)	0.0027 *
*Legionella_var*/LDH	−0.001	0.002	1.0	(0.996; 1.001)	0.39
*Legionella_var*/CRP	0.02	0.004	1.01	(1.008; 1.02)	<0.0001 *
*Legionella_var*/Platelets	−0.0000004	0.000003	1.0	(1.0; 1.0)	0.90
*Legionella_var*/AST	−0.006	0.004	0.99	(0.99; 1.00)	0.08
*Legionella_var*/ALT	−1.12	0.008	1.02	0.999; 1.03)	0.06
*Constant*	−4.64	6.59	―	―	0.87

Note: * = significant test; OR = odds ratios; CI = odds ratios confidence interval at 95%; The null model = −2ln(L_0_), where L_0_ was the likelihood of obtaining the observations if the independent variables did not affect the outcome; the full model: −2ln(L_0_), where L_0_ was the likelihood of obtaining the observations with all independent variables incorporated in the model; C = chi-square test.

## Data Availability

All data analyzed in this study are included in the article.

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
