# Peer review of "Community-Acquired Legionella pneumophila Pneumonia: A Case-Control Cross-Sectional Study in Adult In-Patients from 2019 to 2024"

_pathogens, 2025, doi:10.3390/pathogens15010013_

Round 1

Reviewer 1 Report

Comments and Suggestions for Authors

The authors present a retrospective case-control study conducted between 2019 and 2024 that evaluated clinical, laboratory, and therapeutic characteristics of hospitalized adult patients with pneumonia who tested positive or negative for L. pneumophila. The study includes 140 patients (70 per group), assessed using microbiological diagnostics (urinary antigen and BioFire panel), clinical parameters, comorbidities, risk factors, and treatment choices.

Major concerns

A fundamental question is what new findings this research provides. Did the study only confirm already well-established risk factors for L. pneumophila infection, such as smoking or the presence of comorbidities? The manuscript does not clearly highlight the novelty or unique contribution of the study.

The manuscript does not fully explain the sampling strategy (random selection, consecutive admissions, matching criteria). Mortality is reported but not statistically analyzed or discussed. The study period overlaps with COVID-19 pandemic-related diagnostic changes, yet this factor is neither analyzed nor discussed. In the Discussion section, the authors state that "risk factors as smoking, clinical characteristics, and biochemical parameters, were significantly associated with Legionella-positive cases." However, it is unclear what clinical characteristics the authors refer to. Diagnostic difficulty in legionellosis arises precisely because there are no characteristic clinical symptoms.

There are instances in the manuscript where L. pneumophila is not italicized.

Author Response

Dear Reviewer,

All sections of the manuscript have been improved based on your comments. All the authors thank you for your constructive suggestions, which allowed us to improve the manuscript.

Sincerely,

Nicola Serra

_____________________________________________________________________________________________

REVIEWER 1

The authors present a retrospective case-control study conducted between 2019 and 2024 that evaluated clinical, laboratory, and therapeutic characteristics of hospitalized adult patients with pneumonia who tested positive or negative for L. pneumophila. The study includes 140 patients (70 per group), assessed using microbiological diagnostics (urinary antigen and BioFire panel), clinical parameters, comorbidities, risk factors, and treatment choices.

Major concerns

1) A fundamental question is what new findings this research provides. Did the study only confirm already well-established risk factors for L. pneumophila infection, such as smoking or the presence of comorbidities? The manuscript does not clearly highlight the novelty or unique contribution of the study.

[Reply]: Thank you for your question. We have improved the conclusions section and explained the originality of our study.

2) The manuscript does not fully explain the sampling strategy (random selection, consecutive admissions, matching criteria).

[Reply]: Thank you for your question. in the Methods, we fully explain the sampling strategy (lines 211-218)

3) Mortality is reported but not statistically analyzed or discussed.

[Reply]: Thank you for your question. The mortality was reported and analysed in Table 1(line highlighted in yellow). Additionally, we added in Discussion section the lines:366-372.

4) The study period overlaps with COVID-19 pandemic-related diagnostic changes, yet this factor is neither analyzed nor discussed.

[Reply]: Thank you for your question. This point was not discussed, since among exclusion criteria we do not consider patients positive to HIV and SARS-CoV-2 infection. This was better explained in Methods (lines: 218-219)

5) In the Discussion section, the authors state that "risk factors as smoking, clinical characteristics, and biochemical parameters, were significantly associated with Legionella-positive cases." However, it is unclear what clinical characteristics the authors refer to. Diagnostic difficulty in legionellosis arises precisely because there are no characteristic clinical symptoms.

[Reply]: Thank you for your comment. We changed “clinical characteristics” to “Laboratory parameters (Table 2)”. There was a typo in the final draft of the manuscript.

6) There are instances in the manuscript where L. pneumophila is not italicized.

[Reply]: Thank you for your suggestion. We changed

Reviewer 2 Report

Comments and Suggestions for Authors

Dear authors, 

This manuscript aimed to identify risk factors for LD and clinical outcome among patients with pneumonia hospitalized in one hospital in Italy, using case-control design.

Although the findings are not surprising or absolutely novel, this study took effort for deeper understanding of LD. While I appreciate initiative on this particular subject, I found that the manuscript needs major revision.

All parts of the manuscript need to be revised. Readers would benefit from a clearer organization of information.

Abstract: 

Line 27- LD is a leading cause of HCA pneumonia-please, check the data again

Line 35- explain the abbreviation SG and CG

Introduction

The authors should make clearer what is the gap in the literature that is filled with this study. The study must be actualized by framing it within the vast body of literature that addressed LD. What is the possible international contribution of the study to the literature? What are the implications of the study? The objectives should be better explained. I recommend the authors to enrich the literature review of the paper.

Line 56- reffer precisely why it is important to cut the morbidity and mortality rates worldvide by 2030, add reference please  

Line 60- check the data

Line 67- explain in more details new regional surveillance system in Sicily, Is it related to line 110-113?

Line 109- reference missing

Methods

I miss case definition of LD as well as exclusion criteria. 

Line 120-all enrolled patients with pneumonia was study sample? You mentioned in line 154 hat you randomly selected both groups.   

Was serology use for case definition? One high titer or 4fold increase in paired sera?  

Line 149-  You mentioned that you treat the patients with  clarithromycin and azithromycin and three fluoroquinolones. However, in the table 1 you listed doxycycline, just levofloxacin and  azithromycin. Redefined therapy.

Results:

Clear presentation, but  previous information is needed to understand the results.It would be good to know the settings of infection for cases of Legionnaires’ disease (community acquired, TALD, hospital acquired...)

Discussion: 

I suggest to shorten and refocus discussion. The susceptibility was not among the objectives of the study. Stay with your results. Emphasize the contribution of the study to the literature

Comments on the Quality of English Language

There are many English syntax and spelling issues throughout that make this manuscript difficult and demanding to read. The readers would benefit from a thorough edit by a native English speaker. In addition, one maybe Italian word should be replaced with English one throught the text (ipofosfatemia - hypophosphatemia).

Author Response

Dear Reviewer,

All sections of the manuscript have been improved based on your comments. All the authors thank you for your constructive suggestions, which allowed us to improve the manuscript.

Sincerely,

Nicola Serra

__________________________________________________

REVIEWER 2

Dear authors, 

This manuscript aimed to identify risk factors for LD and clinical outcome among patients with pneumonia hospitalized in one hospital in Italy, using case-control design.

Although the findings are not surprising or absolutely novel, this study took effort for deeper understanding of LD. While I appreciate initiative on this particular subject, I found that the manuscript needs major revision.

All parts of the manuscript need to be revised. Readers would benefit from a clearer organization of information.

Abstract: 

1) Line 27- LD is a leading cause of HCA pneumonia-please, check the data again.

[Reply]: Thank you for your suggestion. We deleted the first sentence, regarding Legionella infection, and changed all Background section. Particularly, we point out that our cases were represented by patients community acquired pneumonia and requiring hospitalization.

2) Line 35- explain the abbreviation SG and CG

[Reply]: Thank you for your question. The explain in abstract was at line 39 and in Method at line 211

Introduction

  • The authors should make clearer what is the gap in the literature that is filled with this study.

[Reply]: Thank you for your question. The paper was improved by more clearly presenting the title, abstract, and all sections of the manuscript to better emphasize our findings for readers. We also added a strengths section to page 10. In addition, our study contribute to added more detailed results on Italian scenario.

  • The study must be actualized by framing it within the vast body of literature that addressed LD.

[Reply]: Thank you for your suggestion. In Introduction and Discussion section more international reference were added. Our results were also compared with those of international studies, such as the mortality rate, laboratory parameters and heart failure.

  • What is the possible international contribution of the study to the literature?

[Reply]: Thank you for your question. The possible international contribution of our study is that our results are not completely in agreement with other studies on this topic. This is likely due to a case-control study design rarely used in the literature. In fact, most of the articles found were retrospective cohort studies.

  • What are the implications of the study?

[Reply]: Thank you for your question. We reported the revised objective and we have improved the conclusions section and explained the originality of our study.

  • The objectives should be better explained. I recommend the authors to enrich the literature review of the paper.

[Reply]: Thank you for your suggestion. The objective was better described, and more references were added in Introduction and Discussion section.

  • Line 56- refer precisely why it is important to cut the morbidity and mortality rates worldvide by 2030, add reference please  

[Reply]: Thank you for your suggestion. We added the reference

      7) Line 60- check the data

[Reply]: Thank you for your suggestion. We have checked and improved

  • Line 67- explain in more details new regional surveillance system in Sicily, Is it related to line 110-113?

[Reply]: Thank you for your question. We added the lines 196-202

Line 109- reference missing

[Reply]: Thank you for your question. We added the reference

Methods

1) I miss case definition of LD as well as exclusion criteria. 

[Reply]: Thank you for your question. We better describe the exclusion criteria in Methods. About the definition of LD we reported in introduction the definition and related references. Lines: 93-97

2) Line 120-all enrolled patients with pneumonia was study sample? You mentioned in line 154 hat you randomly selected both groups.   

[Reply]: Thank you for your question. We better described in Methods the groups and the sampling (lines: 213-218)

3) Was serology use for case definition? One high titer or 4fold increase in paired sera?  

[Reply]: Thank you for your question. Since our study focuses on early diagnosis, serology was included only to demonstrate the thoroughness of the tests performed in our laboratories. For this reason, since this procedure was not considered among those fast and easy, we chose to delete it

4) Line 149 - You mentioned that you treat the patients with clarithromycin and azithromycin and three fluoroquinolones. However, in the table 1 you listed doxycycline, just levofloxacin and azithromycin. Redefined therapy.

[Reply]: Thank you for your question. We improved (lines:200-204)

Results:

1) Clear presentation, but previous information is needed to understand the results. It would be good to know the settings of infection for cases of Legionnaires’ disease (community acquired, TALD, hospital acquired...)

[Reply]: Thank you for your question. We better specify that positive to L. Pneumophila were hospitalized patients with community-acquired pneumonia. (Methods, lines: 211-219)

Discussion: 

1) I suggest to shorten and refocus discussion. The susceptibility was not among the objectives of the study. Stay with your results. Emphasize the contribution of the study to the literature.

[Reply]: Thank you for your suggestion. We refocused the discussion and introduction

Comments on the Quality of English Language

There are many English syntax and spelling issues throughout that make this manuscript difficult and demanding to read. The readers would benefit from a thorough edit by a native English speaker. In addition, one maybe Italian word should be replaced with English one throught the text (ipofosfatemia - hypophosphatemia).

[Reply]: Thank you for your suggestion. The English language was revised 

Round 2

Reviewer 1 Report

Comments and Suggestions for Authors

Please ensure that L. pneumophila is italicized throughout the entire manuscript (e.g., italics are missing in lines 398–399).

Reviewer 2 Report

Comments and Suggestions for Authors

The authors responded well.